# Dissolution Thermodynamics of the Solubility of Sulfamethazine in (Acetonitrile + 1-Propanol) Mixtures

**DOI:** 10.3390/ph17121594

**Published:** 2024-11-26

**Authors:** Daniel Ricardo Delgado, Jennifer Katiusca Castro-Camacho, Claudia Patricia Ortiz, Diego Ivan Caviedes-Rubio, Fleming Martinez

**Affiliations:** 1Programa de Ingeniería Civil, Grupo de Investigación de Ingenierías UCC-Neiva, Facultad de Ingeniería, Universidad Cooperativa de Colombia, Sede Neiva, Calle 11 No. 1-51, Neiva 410001, Huila, Colombia; diego.caviedesr@campusucc.edu.co; 2Programa de Ingeniería Agroindustrial, Hidroingeniería y Desarrollo Agropecuario, Facultad de Ingeniería, Universidad Surcolombiana, Neiva 410001, Huila, Colombia; jenniferkatiusca.castro@usco.edu.co; 3Programa de Administración en Seguridad y Salud en el Trabajo, Grupo de Investigación en Seguridad y Salud en el Trabajo, Corporación Universitaria Minuto de Dios-UNIMINUTO, Neiva 410001, Huila, Colombia; claudia.ortiz.de@uniminuto.edu.co; 4Grupo de Investigaciones Farmacéutico-Fisicoquímicas, Departamento de Farmacia, Facultad de Ciencias, Universidad Nacional de Colombia, Sede Bogotá, Carrera 30 No. 45-03, Bogotá 110321, Cundinamarca, Colombia

**Keywords:** sulfamethazine, solubility, acetonitrile, 1-propanol, cosolvency, solution thermodynamics

## Abstract

**Background**: Solubility is one of the most important parameters in the research and development processes of the pharmaceutical industry. In this context, cosolubility is one of the most used strategies to improve the solubility of poorly soluble drugs, besides allowing to identify some factors involved in the dissolution process. The aim of this research is to evaluate the solubility of sulfamethazine in acetotinitrile + 1-propanol cosolvent mixtures at 9 temperatures (278.15, 283.15, 288.15, 293.15, 298.15, 303.15, 308.15, 313.15, and 318.15 K); a drug used in human and veterinary therapy and two solvents of great chemical–pharmaceutical interest. **Methods**: The determination was carried out by the shaking flask method and the drug was quantified by UV/Vis spectrophotometry. **Results**: The solubility of sulfamethazine increases from pure 1-propanol (solvent in which it reaches its lowest solubility at 278.15 K) to pure acetonitrile (solvent in which it reaches its maximum solubility at 318.15 K), behaving in a logarithmic-linear fashion. **Conclusions**: The increase in solubility is related to the acid/base character of the cosolvent mixtures and not to the solubility parameter of the mixtures. The dissolution process is endothermic and favored by the solution entropy, and also shows a strong entropic compensation.

## 1. Introduction

Sulfamethazine (SMT) (Figure 1, IUPAC: 4-amino-*N*-(4,6-dimethylpyrimidin-2-yl) benzenesulfonamide; Molecular Formula C_12_H_14_N_4_O_2_S) is a broad-spectrum bacteriostatic antimicrobial used in veterinary medicine to treat infections caused by some bacteria in the pasteurellaceae, bacteroidacecae, enterobacterias, morganellaceae, and bacillota families. It is also used to treat infections of the respiratory, digestive, genitourinary, central nervous, and musculoskeletal systems, and in human therapy to treat bacterial infections such as bronchitis, prostatitis, and urinary tract infections [1,2,3,4,5,6].

The efficacy of this drug has made SMT the main treatment option for infections, particularly in veterinary medicine, which has led to an increase in its use, resulting in higher levels of spillage that can eventually reach soil or surface water, making SMT an environmental concern, so the SMT is considered as an emerging contaminant of pharmaceutical origin, classified by the NORMAN network as Category I [7], which in turn has led to the development of a large number of decontamination methods [8].

**Figure 1 pharmaceuticals-17-01594-f001:**
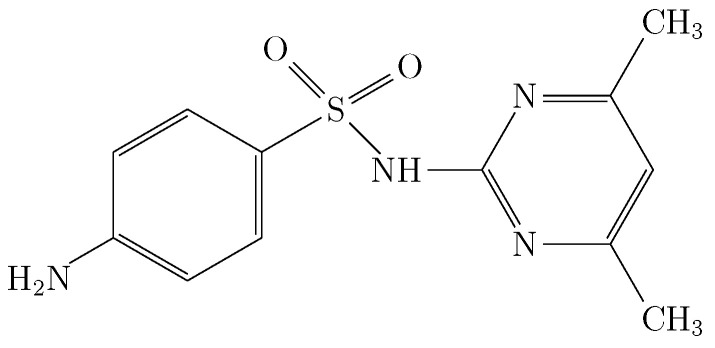
Molecular structure of the 4-amino-*N*-(4,6-dimethylpyrimidin-2-yl)benzenesulfonamide [9].

In relation to the above, the pharmaceutical industry must promote the development of more efficient processes and drugs. In this context, solubility studies play an important role as solubility is closely related to pre-formulation, formulation, purification, recrystallization, quantification, quality control, dosage, and other processes. One of the best known strategies for improving or controlling the solubility of Active Pharmaceutical Ingredients (APIs) is cosolubility, which is the mixing of two or more solvents to improve the solubility of an API [10,11,12].

Given the importance of solubility, the development of studies that not only generate relevant data for the pharmaceutical industry, but also allow understanding the possible molecular interactions between the drug and the solvents, generating information that contributes to the development of products or processes in a more rational way. In this context, the present study contributes to the understanding of the influence of structural changes of some sulfonamides in structurally related organic solvents and cosolvent mixtures of pharmaceutical interest. Thus, in this work, we report the solubility of SMT in cosolvent mixtures of acetonitrile (1) + 1-propanol (2), two solvents of great interest for the chemical industry, especially for the pharmaceutical industry.

Acetonitrile (MeCN) is an essential solvent in the chemical industry, especially in the pharmaceutical industry, where its global demand exceeds 70% of the total market [13,14]. This popularity is due to its excellent solvating capacity for a wide variety of drugs of different polarities. Other important properties that make MeCN an important solvent in the pharmaceutical industry are its low freezing and boiling points, low viscosity, and relatively low toxicity. Two of the most important uses of MeCN are: (a) its use in laboratories, particularly as a mobile phase in liquid chromatographic analysis techniques [15], and (b) as a solvent in industrial processes, such as the manufacture of antibiotics and other pharmaceuticals [13]. On the other hand, 1-propanol (1-PrOH) is used as an extraction medium for natural products, such as aromatic substances, oils of plant origin, resins, waxes, and gums [16]. In addition to its solvent properties, 1-PrOH is used in pharmaceuticals and cosmetics for its antiseptic properties. An important application in the chemical industry is its use as a chemical intermediate in the production of compounds of pharmaceutical interest [17,18,19].

The aim of this work is, therefore, to present the thermodynamic analysis of the solubility of SMT in cosolvent mixtures (MeCN + 1-PrOH). This study allows further analysis of the relationship between solubility and the chemical structure of the drug and some linear alcohols (methanol (MeOH), ethanol (EtOH), and 1-propanol (1-PrOH)). This work is the continuation of a series of studies on the solubility of three sulfonamides (sulfadiazine, sulfamerazine, and SMT) in three cosolvent systems {MeCN (1) and MeOH (2)} [20,21,22], {MeCN (1) + EtOH (2)} [23], and {MeCN (1) + 1-PrOH (2)} [24,25]. In particular, this study shows that while the solubility of these three drugs in aqueous systems is strongly related to the solubility parameter of the cosolvent mixture [26,27,28,29,30,31], in purely organic cosolvent systems (MeCN + alcohol), this behaviour is partly present in the MeCN (1) + MeOH (2) system.

## 2. Results and Discussion

### 2.1. Experimental Mole Fraction Solubility

The solubility expressed as mole fraction ($x_{3}$) of SMR in pure solvents and cosolvent mixtures is calculated using Equation (Equation 1):
(1)$$x_{3}=\frac{m_{3}/M_{3}}{m_{1}/M_{1}+m_{2}/M_{2}+m_{3}/M_{3}}$$
where $m_{1}$, $m_{2}$, and $m_{3}$ are the masses of MeCN, 1-PrOH, and SMT, respectively, and $M_{1}$, $M_{2}$, and $M_{3}$ are the molar masses of MeCN, 1-PrOH, and SMT, respectively.

Table 1 shows the solubility of SMT as a function of the MeCN mass fraction ($w_{1}$), specifically in 19 cosolvent mixtures {MeCN (1) + 1-PrOH (2)}, (from $w_{1}$ = 0.05 to $w_{1}$ = 0.95) and 2 pure solvents (MeCN and 1-PrOH) at 9 temperatures (from 278.15 to 318.15 K). When analyzing the effect of MeCN addition in cosolvent mixtures on the behavior of the mole fraction ($x_{3}$), of SMT, the solubility increases between 5.6 and 7.2 times from pure 1-PrOH to pure MeCN increases between 5.6 and 7.2 times from pure 1-PrOH to pure MeCN, showing a positive cosolvent effect of MeCN and an antisolvent effect of 1-PrOH (Figure 2A). On the other hand, when the solubility is evaluated as a function of temperature, the solubility increases with temperature in all cases, indicating an endothermic process (Figure 2B). In pure MeCN, the solubility increases about 3.6-fold with increasing temperature from 278.15 to 318.15 K, and in pure 1-PrOH, the increase is 4.7-fold in the same temperature range.

**Table 1 pharmaceuticals-17-01594-t001:** Experimental mole fraction solubility ($10^{4}\cdot x_{3}$) of SMT (3) in {MeCN (1) + 1-PrOH (2)} mixtures at several temperatures (in K) and *p*/kPa = 100 MPa ^*a*^.

*w*_1_ ^*a,b*^	Temperature/K *^b^*								
	**278.15**	**283.15**	**288.15**	**293.15**	**298.15**	**303.15**	**308.15**	**313.15**	**318.15**
0.00	3.17	3.75	4.42	5.32	6.18	7.08	8.40	10.1	11.5
0.05	3.51	4.14	4.93	5.95	7.01	7.94	9.46	11.3	13.0
0.10	3.82	4.52	5.40	6.50	7.65	8.65	10.4	12.4	14.3
0.15	4.20	4.97	5.87	7.13	8.16	9.56	11.4	13.7	15.9
0.20	4.54	5.46	6.49	7.82	9.15	10.6	12.5	15.1	17.4
0.25	4.93	5.93	7.06	8.47	9.81	11.7	13.9	16.6	19.3
0.30	5.42	6.45	7.69	9.28	10.8	12.8	15.3	18.4	21.2
0.35	5.89	7.08	8.49	10.3	12.1	14.2	16.8	20.2	23.4
0.40	6.45	7.72	9.28	11.1	13.4	15.8	18.5	22.4	25.9
0.45	6.96	8.51	10.1	12.2	14.7	17.2	20.4	24.5	28.5
0.50	7.69	9.15	11.2	13.4	16.2	19.0	22.4	26.9	31.4
0.55	8.31	10.1	12.2	14.9	17.6	21.0	24.7	29.7	35.0
0.60	9.05	11.0	13.4	16.1	19.1	23.6	27.3	32.8	38.4
0.65	9.91	12.0	14.5	17.3	20.9	25.6	30.0	36.0	42.2
0.70	10.8	13.2	16.1	19.1	23.3	28.4	33.2	40.0	46.6
0.75	11.7	14.4	17.6	21.0	25.6	30.1	36.0	43.7	51.5
0.80	13.0	15.6	19.0	23.6	28.2	34.1	40.4	48.0	57.1
0.85	14.0	17.1	20.8	25.7	30.5	36.9	44.1	53.1	62.8
0.90	15.1	18.8	23.0	27.8	33.5	40.5	48.9	58.5	68.3
0.95	16.7	20.6	25.2	30.6	37.3	45.2	53.1	64.9	76.3
1.00 ^c^	17.8	22.0	27.1	33.1	40.2	48.8	58.1	70.0	83.1
Ideal ^c^	101.4	118.4	137.9	160.1	185.5	214.4	247.2	284.3	326.3

^*a*^ $w_{1}$ is the mass fraction of MeCN (1) in the {MeCN (1) + 1-PrOH (2)} mixtures free of SMT (3). ^*b*^ Average relative standard uncertainty in $w_{1}$ is $u_{r}$($w_{1}$) = 0.0010. Standard uncertainty in *T* is *u*(*T*) = 0.10 K. Average relative standard uncertainties in $x_{3}$ is $u_{r}$($x_{3(1+2)}$) = 0.020. ^*c*^ Data from Ortiz et al. [22].

The trends shown in Figure 2 were fitted to linear mathematical models. Thus, the log solubility was correlated by a linear bivariate model (Equation (Equation 1)), where $w_{1}$ is the mass fraction of 1-PrOH free of SMT and *T* is the temperature expressed in Kelvin (K). With respect to the model, the adjusted coefficient of determination $r^{2}$ is very close to 1 (0.999), the standard error is 0.028, and the *F*-statistic value is 63499, indicating that the results are statistically significant. When the solubility of SMT is calculated using the model (Equation (Equation 2)), the data show a mean percentage deviation (MPD%) of 0.34% (Equation (Equation 3)—with *n* = 189); when the direct solubility of SMT is calculated (Equation (Equation 4)—with *n*= 189), the MPD% is 2.23%.
(2)$$\ln x_{3}=17.96\left(\pm 0.05\right)+1.864\left(\pm 0.007\right)\cdot w_{1}+3.547\left(\pm 0.016\right)\cdot 10^{-2}\cdot T$$
(3)$$\mathrm{MPD\%}=\sum_{i=1}^{n}\left|\left(\ln x_{3}^{\mathrm{calculated}}-\ln x_{3}^{\mathrm{observed}}\right)\cdot {\left(\ln x_{3}^{\mathrm{observed}}\right)}^{-1}\right|\cdot n^{-1}$$
(4)$$\mathrm{MPD\%}=\sum_{i=1}^{n}\left|\left(x_{3}^{\mathrm{calculated}}-x_{3}^{\mathrm{observed}}\right)\cdot {\left(x_{3}^{\mathrm{observed}}\right)}^{-1}\right|\cdot n^{-1}$$

On the other hand, Figure 3 allows the comparison of SMT solubility in {MeCN (1) + 1-PrOH (2)} mixtures with respect to its structural analogs sulfadiazine (SD) and sulfamerazine (SMR) (Figure 1) at *T*/K = 298.15 [24,25]. As observed, SMT solubility values are the highest in all cases, but SD values are the lowest. This behavior is also observed in aqueous mixtures (MeOH + water, EtOH + water, and 1-PrOH + water), where the order of solubility is SMT > SMR > SD in all three cosolvent systems. This behavior may be due to characteristics of the crystal structure of each sulfonamide, which can be compared by temperature and/or enthalpy of fusion: SD ($T_{\mathrm{fus}}=532.65$ K; $\Delta_{\mathrm{fus}}H=44.3$ kJ·mol^−1^) > SMR ($T_{\mathrm{fus}}=508.30$ K; $\Delta_{\mathrm{fus}}H=41.3$ kJ·mol^−1^) > SMT ($T_{\mathrm{fus}}=471.65$ K; $\Delta_{\mathrm{fus}}H=39.2$ kJ·mol^−1^) [32]. As can be seen, the intermolecular forces are highest for SD, which has the lowest solubility, and lowest for SMT, which has the highest solubility at the same temperature and solvent conditions.

**Figure 3 pharmaceuticals-17-01594-f003:**
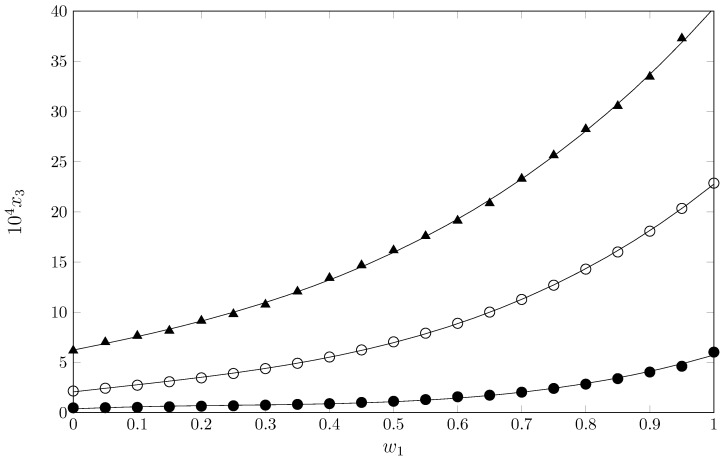
Mole fraction solubility of SD (3, •) [24], SMR (3, ∘) [25], and SMT (3, ▲) [this work] in {MeCN (1) + 1-PrOH (2)} mixtures at *T*/K = 298.15.

Figure 4 shows the solubility of SMT in the cosolvent mixtures {MeCN (1) + MeOH (2)} [22] and {MeCN (1) + 1-PrOH (2)} at 298.15 K as a function of the cosolvent composition (top) and the solubility parameter of the cosolvent mixtures (bottom). It is usually expected that maximum solubility will be achieved in a solvent or cosolvent mixture with a solubility parameter similar or equal to that of the drug [33]; thus, the solubility parameter of SMT is 27.42 MPa^1/2^, for MeOH it is 29.6 MPa^1/2^ [34,35], for 1-PrOH it is 24.6 MPa^1/2^ [34,36], and for MeCN it is 24.4 MPa^1/2^ [34,37], and therefore, it would be expected that the maximum solubility would be obtained in MeOH, which is the solvent with a similar solubility parameter to that of SMT (93% similarity), but as shown in Figure 3, SMT is less soluble in this solvent.

**Figure 4 pharmaceuticals-17-01594-f004:**
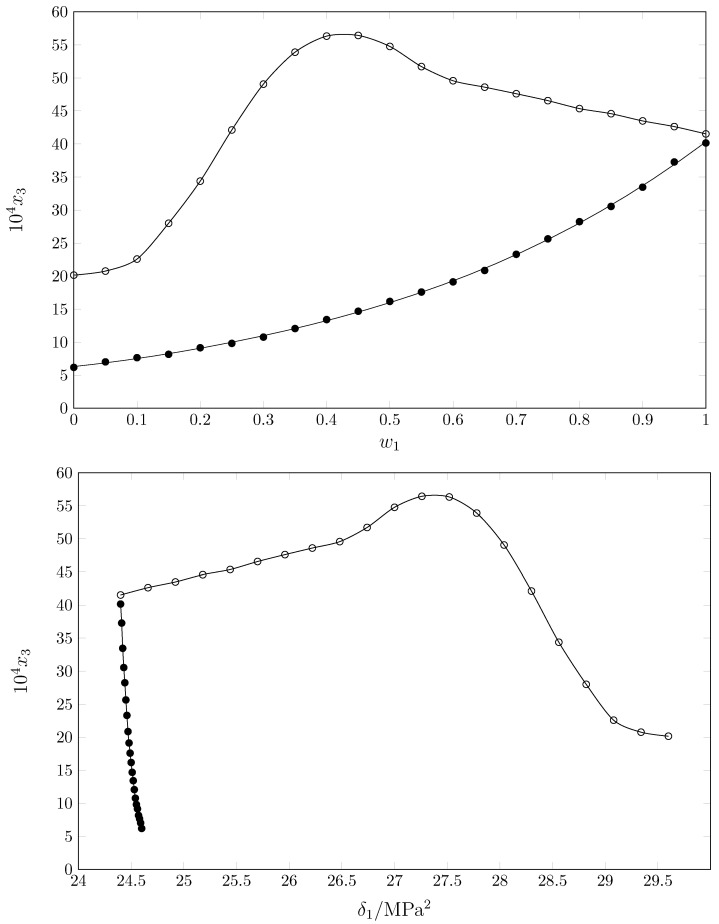
Mole fraction solubility of SMT (3) as a function of mixtures’ composition (top) and as a function of the Hildebrand solubility parameter of the mixtures (bottom) in some {MeCN (1) + alcohol (2)} mixtures at *T*/K = 298.15. ∘: {MeCN (1) + MeOH (1)} [22]; •: {MeCN (1) + 1-PrOH (2)} [this work].

In the {MeCN (1) + MeOH (2)} system, the Hildebrand principle is partially fulfilled, since the maximum solubility is reached in a cosolvent mixture with a solubility parameter similar to that of SMT (approximately 27 MPa^1/2^), but the solubility in MeCN is approximately twice the solubility in MeOH. When evaluating the solubility behaviour in the {MeCN (1) + 1-PrOH (2)} system, where the solubility parameter is similar in all the mixtures (Δδ = 0.2 MPa^1/2^), the maximum solubility is reached in MeCN, so the solubility profile tends to be related to the acid/base character of the solvents, and thus, SMT is more soluble in basic media ($\alpha_{MeCN}$ = 0.29 ± 0.006) and less soluble in more acidic solvents. It is observed that the increase in solubility is inversely related to the acidic character of the solvent ($\alpha_{MeCN}$ = 0.2, $\alpha_{1-PrOH}$ = 0.766, $\alpha_{MeOH}$ = 0.990), which would explain the lower solubility of SMT in MeOH compared to 1-PrOH and MeCN, and the increase in solubility upon addition of MeCN; the basic character of the mixture would increase, favoring the solubility of SMT.

### 2.2. SMT Solid Phases DSC Analysis

The SMT DSC thermograms of the untreated commercial sample and of the solid phases in equilibrium with the saturated solutions in the pure solvents MeCN and 1-PrOH and the mixture $w_{1}$ = 0.50 are shown in Figure 5. In all cases, there is an endothermic peak corresponding to the melting of the SMT. The melting temperature varied by 0.4 degrees ($T_{\mathrm{fus}}$/K = from 468.3 K to 468.7 K), which is very similar to the data reported in the literature for the original untreated sample ($T_{\mathrm{fus}}$/K = 468.9) [38]. In general, no polymorphic transformations or solvate formation were observed after the saturation process.

### 2.3. Activity Coefficients of SMT in Mixed Solvents

On an asymmetric basis, the SMT activity coefficients ($\gamma_{3}$) of the ideal and experimental SMT solubilities shown in Table 2 have been calculated as the quotient $x_{3}^{\mathrm{id}}/x_{3}$. The lowest and highest SMT solubilities are found at *T*/K = 298.15, where the $\gamma_{3}$ values vary from 30.0 in neat 1-PrOH to 4.4 in neat MeCN. Note that the $\gamma_{3}$ values decrease with increasing temperature in every solvent system examined. From the SMT $\gamma_{3}$ values, a good approximation of the SMT-solvent intermolecular interactions can be obtained using the following equation [39]:(5)$$\ln\gamma_{3}=\left(e_{ss}+e_{33}-2e_{s3}\right)\frac{V_{3}\phi_{s}^{2}}{RT}$$

The solvent is denoted here by the subscript, which stands for the neat solvents MeCN and 1-PrOH or the {MeCN (1) + 1-PrOH (2)} mixtures. The solvent–solvent, SMT–SMT and solvent–SMT interaction energies are represented by $e_{ss}$, $e_{33}$, and $e_{s3}$, respectively; $V_{3}$ is the molar volume of the supercooled liquid SMT, while $\phi_{s}$ is the volumetric fraction of the solvent system. Regardless of the solvent system, the expression $V_{3}\phi_{s}^{2}/RT$ can be considered constant for low values of $x_{3}$. Accordingly, $e_{ss}$, $e_{33}$, and $e_{s3}$ would be the primary determinants of the SMT $\gamma_{3}$ [39]. The $e_{s3}$ term favors the corresponding SMT dissolution processes and is solubility-increasing, while the $e_{ss}$ and $e_{s3}$ terms are detrimental to SMT solubility and dissolution. The contribution of the $e_{33}$ term could be considered as constant and independent of the solvent systems.

**Table 2 pharmaceuticals-17-01594-t002:** Activity coefficients of SMT (3) in {MeCN (1) + 1-PrOH (2)} mixtures at several temperatures (in K).

$w_{1}^{a}$	Temperature/K ^*bc*^								
	**278.15**	**283.15**	**288.15**	**293.15**	**298.15**	**303.15**	**308.15**	**313.15**	**318.15**
0.00	32.0	31.5	31.2	30.1	30.0	30.3	29.4	28.1	28.3
0.05	28.9	28.6	28.0	26.9	26.5	27.0	26.1	25.2	25.1
0.10	26.5	26.2	25.6	24.6	24.2	24.8	23.8	23.0	22.8
0.15	24.2	23.8	23.5	22.5	22.7	22.4	21.6	20.7	20.5
0.20	22.3	21.7	21.2	20.5	20.3	20.3	19.7	18.8	18.7
0.25	20.6	20.0	19.5	18.9	18.9	18.3	17.7	17.1	16.9
0.30	18.7	18.4	17.9	17.3	17.2	16.7	16.2	15.5	15.4
0.35	17.2	16.7	16.3	15.6	15.4	15.1	14.7	14.0	13.9
0.40	15.7	15.3	14.9	14.5	13.8	13.6	13.4	12.7	12.6
0.45	14.6	13.9	13.6	13.1	12.6	12.5	12.1	11.6	11.4
0.50	13.2	12.9	12.3	12.0	11.5	11.3	11.0	10.6	10.4
0.55	12.2	11.8	11.3	10.8	10.6	10.2	10.0	9.6	9.3
0.60	11.2	10.8	10.3	10.0	9.7	9.1	9.0	8.7	8.5
0.65	10.2	9.8	9.5	9.3	8.9	8.4	8.2	7.9	7.7
0.70	9.4	9.0	8.6	8.4	8.0	7.5	7.4	7.1	7.0
0.75	8.7	8.2	7.8	7.6	7.2	7.1	6.9	6.5	6.3
0.80	7.8	7.6	7.3	6.8	6.6	6.3	6.1	5.9	5.7
0.85	7.2	6.9	6.6	6.2	6.1	5.8	5.6	5.4	5.2
0.90	6.7	6.3	6.0	5.8	5.5	5.3	5.1	4.9	4.8
0.95	6.1	5.8	5.5	5.2	5.0	4.7	4.7	4.4	4.3
1.00 ^c^	5.7	5.4	5.1	4.8	4.6	4.4	4.3	4.1	3.9

^*a*^ $w_{1}$ is the mass fraction of MeCN (1) in the {MeCN (1) + 1-PrOH (2)} mixtures free of SMT (3). ^*b*^ Average relative standard uncertainty in $w_{1}$ is $u_{r}$($w_{1}$) = 0.0010. Standard uncertainty in *T* is *u*(*T*) = 0.10 K. Average relative uncertainty in $\gamma_{3}$ is $u_{r}$($\gamma_{3}$) = 0.030. ^*c*^ Data from Ortiz et al. [22].

Based on the energy quantities given in Equation (Equation 5), the following qualitative method could be carried out: Compared to MeCN (1) ($\delta_{1}$ = 24.1 MPa^1/2^) [34,40], the $e_{ss}$ is slightly larger in clean 1-PrOH (2) ($\delta_{2}$ = 24.4 MPa^1/2^). While $e_{ss}$ values near 5.0 are significantly lower and $e_{s3}$ values are significantly higher in pure MeCN, pure 1-PrOH with SMT $\gamma_{3}$ values near 30 would infer moderately high $e_{ss}$ and relatively low $e_{s3}$ values. So, it makes sense that neat MeCN would have a higher SMT solvation than 1-PrOH.

### 2.4. Apparent Dissolution Thermodynamics

For the SMT dissolution processes, all apparent thermodynamic quantities were calculated at the harmonic mean temperature ($T_{\mathrm{hm}}$/K = 297.6), which was determined using Equation (Equation 6) [41,42]:(6)$$T_{\mathrm{hm}}=\frac{n}{\sum_{i=1}^{n}\left(\frac{1}{T_{i}}\right)}$$
where the number of temperatures studied is *n* = 9. Therefore, using the statistically modified va not Hoff equation as shown in Equation (Equation 7), the apparent standard molar enthalpy changes of dissolution ($\Delta_{\mathrm{soln}}H^{\circ }$) were determined [23,41]:(7)$$\Delta_{\mathrm{soln}}H^{\circ }=-R{\left(\frac{\partial \ln x_{3}}{\partial \left(T^{-1}-T_{\mathrm{hm}}^{-1}\right)}\right)}_{p}$$

Equation (Equation 8) was used to determine the apparent standard molar Gibbs energy changes for the dissolution processes ($\Delta_{\mathrm{soln}}G^{\circ }$) [23,41]:(8)$$\Delta_{\mathrm{soln}}G^{\circ }=-RT_{\mathrm{hm}}\cdot \mathrm{intercept}$$
which, as shown in Figure 6, used the intercepts from the linear regressions of $\ln x_{3}$ as a function of ($T^{-1}-T_{\mathrm{hm}}^{-1}$). In each regression, linear trends with $r^{2}$ values greater than 0.998 were found [43,44,45]. Finally, Equation (Equation 9) was used to determine the apparent standard molar entropy changes for the dissolution processes ($\Delta_{\mathrm{soln}}S^{\circ }$) from the corresponding $\Delta_{\mathrm{soln}}H^{\circ }$ and $\Delta_{\mathrm{soln}}G^{\circ }$ values:(9)$$\Delta_{\mathrm{soln}}S^{\circ }=\left(\Delta_{\mathrm{soln}}H^{\circ }-\Delta_{\mathrm{soln}}G^{\circ }\right)T_{\mathrm{hm}}^{-1}$$

The values of the solution thermodynamic functions of the SMT in pure MeCN, pure 1-PrOH, and the cosolvent mixtures {MeCN (1) + 1-PrOH (2)} at $T_{\mathrm{hm}}$/K = 297.6 are shown in Table 3.

In all solvent systems, the corresponding apparent molar dissolution enthalpies and entropies, as well as the apparent standard molar Gibbs energies of dissolution of SMT, are positive. Consequently, endothermic and entropy-driven global SMT dissolution processes are always present. The highest and lowest $\Delta_{\mathrm{soln}}G^{\circ }$ values were found in neat 1-PrOH, where the solubilities were at their lowest, and in neat MeCN, where the solubilities were at their maximum. To reach the maximum in neat MeCN, the values of $\Delta_{\mathrm{soln}}H^{\circ }$ and $\Delta_{\mathrm{soln}}S^{\circ }$ increase steadily from neat 1-PrOH.

In addition, using Equations (Equation 10) and (Equation 11) [46], the relative contributions of enthalpy ($\zeta_{H}$) and entropy ($\zeta_{TS}$) to the SMT dissolution processes were determined. The positive enthalpy ($\zeta_{H}$> 0.65), which shows the energetic preponderance in all these SMT dissolution processes, was the main contributor to the positive standard molar Gibbs energies of SMT dissolution, as shown in Table 3.
(10)$$\zeta_{H}=|\Delta_{\mathrm{soln}}H^{\circ }|(|T\Delta_{\mathrm{soln}}S^{\circ }\left|+\right|\Delta_{\mathrm{soln}}H^{\circ }{\left|\right)}^{-1}$$
(11)$$\zeta_{TS}=1-\zeta_{H}$$

### 2.5. Apparent Mixing Thermodynamics

The following hypothetical steps could be used to depict the global dissolution processes of SMT in {MeCN (1) + 1-PrOH (2)} solvent systems from a broad thermodynamic perspective.

Solute_(Solid)_ at $T_{\mathrm{hm}}$→ Solute_(Solid)_ at $T_{\mathrm{fus}}$→ Solute_(Liquid)_ at $T_{\mathrm{fus}}$→ Solute_(Liquid)_ at $T_{\mathrm{hm}}$→ Solute_(Solution)_ at $T_{\mathrm{hm}}$, where (i) the SMT is heated and melted, (ii) the liquid drug is cooled to the study temperature ($T_{\mathrm{fus}}$/K = 297.6), and (iii) the solvent mixture and the hypothetical supercooled liquid drug are then mixed at $T_{\mathrm{fus}}$/K = 297.6 [47,48]. Equations (Equation 12) and (Equation 13) allow the calculation of the different thermodynamic contributions to the global dissolution process as a result of this treatment:(12)$$\Delta_{\mathrm{soln}}H^{o}=\Delta_{\mathrm{mix}}H^{o}+\Delta_{\mathrm{fus}}H^{T_{\mathrm{hm}}}$$
(13)$$\Delta_{\mathrm{soln}}S^{o}=\Delta_{\mathrm{mix}}H^{o}+\Delta_{\mathrm{fus}}S^{T_{\mathrm{hm}}}$$
where the thermodynamic quantities of the SMT (3) fusion and its cooling at $T_{\mathrm{hm}}$/K = 297.6 are given by $\Delta_{\mathrm{fus}}H^{T_{\mathrm{hm}}}$ and $\Delta_{\mathrm{fus}}S^{T_{\mathrm{hm}}}$. The enthalpy and entropy values for the ideal dissolution process, $\Delta_{\mathrm{soln}}H^{o-\mathrm{id}}$ and $\Delta_{\mathrm{soln}}S^{o-\mathrm{id}}$, given in Table 3, are equivalent to these fusion quantities. The mixing of supercooled liquid SMT with all MeCN (1) + 1-PrOH (2) mixtures and with the pure solvents, MeCN and 1-PrOH, at $T_{\mathrm{hm}}$/K = 297.6 is summarized in Table 4. Since the SMT experimental solubilities are smaller than the ideal solubilities, the Gibbs energy of mixing is always positive (Table 1).

Depending on the nature of the mixtures, the contributions of the mixing process molar amounts to the global dissolution processes of SMT are either positive or negative for $\Delta_{\mathrm{mix}}H^{\circ }$, but they are always positive for $\Delta_{\mathrm{mix}}S^{\circ }$. In this sense, $\Delta_{\mathrm{mix}}S^{\circ }$ is positive in MeCN-rich mixtures and in pure MeCN, but negative from pure 1-PrOH to the mixture of $w_{1}$ = 0.65. Thus, in mixtures of 0.70 $\leq w_{1}\leq$ 1.00 the mixing processes are entropy driven due to the entropy-increasing nature of these solvent systems. Due to the negentropic mixing tendency, neither enthalpy nor entropy driving is seen in mixtures of 0.00 $\leq w_{1}\leq$ 0.65.

### 2.6. Enthalpy–Entropy Compensation (EEC) Analysis

In a series of physicochemical studies of the solubility of numerous drugs in various mixed aqueous and non-aqueous cosolvent systems, non-enthalpy–entropy compensatory effects were found [49,50,51]. In particular, based on the composition of the mixtures, these investigations were conducted to determine the primary molecular mechanisms underlying the drug dissolution and/or transfer cosolvent action. Weighted graphs of $\Delta_{\mathrm{soln}}H^{\circ }$ vs. $\Delta_{\mathrm{soln}}G^{\circ }$ (Figure 7 (top)) allowed such an analysis [52,53]. Moreover, plots of $\Delta_{\mathrm{soln}}H^{\circ }$ vs. $T\Delta_{\mathrm{soln}}S^{\circ }$ could also be used [54]. As shown in Figure 7 (top), SMT exhibits a linear $\Delta_{\mathrm{soln}}H^{\circ }$ vs. $\Delta_{\mathrm{soln}}G^{\circ }$ trend with negative slope as described by Equation (Equation 14), with adjusted $r^{2}$ = 0.997, typical error = 0.075 and statistical *F* value = 6813 [43,44]. Thus, this continuous negative slope implies that the driving mechanism for the SMT transfer from the most polar solvent system (namely, neat 1-PrOH) to a less polar solvent (namely, neat MeCN) is entropy-increasing, owing to the possible breaking of hydrogen bonding in the inner neat 1-PrOH structure [40]. Moreover, as shown in Figure 7 (bottom), SMT also exhibits a linear $\Delta_{\mathrm{soln}}H^{\circ }$ vs. $T\Delta_{\mathrm{soln}}S^{\circ }$ trend with positive slope as described by Equation (Equation 15), with adjusted $r^{2}$ = 0.999, typical error = 0.038, and statistical *F* value = 26325 [34]. Entropy driving for SMT transfer from 1 to PrOH to MeCN is also observed because the positive slope of this last plot is lower than the unit.
(14)$$\Delta_{\mathrm{soln}}H^{\circ }=41.49\left(\pm 0.19\right)-0.963\left(\pm 0.012\right)\Delta_{\mathrm{soln}}G^{\circ }$$
(15)$$\Delta_{\mathrm{soln}}H^{\circ }=21.13\left(\pm 0.03\right)+0.491\left(\pm 0.003\right)T\Delta_{\mathrm{soln}}S^{\circ }$$

## 3. Materials and Methods

### 3.1. Reagents

Table 5 lists all the chemicals, drugs, and solvents used in research. Reagents were used in their original condition without further purification.

### 3.2. Solubility Determination

Solubility was determined using the shake flask method according to Higuchi and Connors [55,56,57]. This method has five steps:Saturation: First, approximately 10 mL of each of the cosolvent mixtures (in this case, {MeCN (1) + 1-PrOH (2)} mixtures) is added to a 15 mL amber glass bottle. Small amounts of drug are then added until two phases (liquid/solid) are obtained. This process is performed at room temperature.Thermodynamic equilibrium: Once the solvents or cosolvent mixtures are saturated, the samples are placed in a circulating bath (Medingen K-22/T100, Medingen, Germany), where they are kept at a constant temperature (study temperatures: 278.15–318.15 K) for at least 48 h. During the first 36 h, the samples are shaken continuously to ensure the presence of two phases (solid–solution). After 24 h, the concentration of the solution is determined every 4 h until a constant concentration is obtained to ensure thermodynamic equilibrium between the solid phase and the solution. The sample is then allowed to rest for at least 12 h [58]. Since it is necessary to evaluate the solid phase at equilibrium in order to assess possible changes in the crystalline structure of the SMT that may affect solubility, the samples are kept at a temperature 5 degrees above the study temperature for the first 8 h to ensure a saturated solution and to obtain crystals by precipitation with decreasing temperature, necessary for step 5 of the shake flask method.Filtration: To ensure that the solution was free of solid particles at the time of dilution to determine the concentration of each sample, each sample was filtered through a 0.45 μm membrane (Millipore Corp. Swinnex-13, Burlington, MA, USA). To minimize concentration changes due to temperature changes at the time of filtration, syringes and filters are brought to the study temperature.Quantification: The quantification of SMT is performed by UV/Vis spectrophotometry (UV/Vis EMC-11- UV spectrophotometer, Duisburg, Germany). A calibration curve is developed in absolute ethanol due to the good solubility of the drug in this solvent, which guarantees the respect of the Lambert-Beer law; all the dilutions necessary for the quantification of SMT are made in absolute ethanol to avoid possible precipitations. Since the dilutions required for quantification are in most cases greater than 1:100, the effect of the solvents MeCN and 1-PrOH on the linearity or wavelength of maximum absorbance is negligible.Evaluation of the solid phase: To evaluate possible polymorphic changes or decomposition of SMR, the solid phases in equilibrium with the saturated solutions are analyzed by DSC (DSC 204 F1 Phoenix, Dresden, Germany). Sample mass: approximately 10.0 mg; calibration samples: indium and tin as standards; nitrogen flow: 10 mL·min^−1^; heating ramp: 10 K·min^−1^.

## 4. Conclusions

According to the solubility results of SMT (3) in cosolvent mixtures {MeCN (1) + 1-PrOH (2)}, the dissolution process is endothermic and strongly influenced by the composition of the cosolvent. Comparing the results with some studies in aqueous cosolvent mixtures, a possible change in the mechanisms involved in the dissolution process is identified. Thus, in aqueous systems, the solubility of SMT is influenced by the solubility parameter of the mixture, but in the system {MeCN (1) + 1-PrOH (2)}, an opposite behavior is observed, identifying a possible relationship between the solubility and the acid/base character of the cosolvent mixture. Regarding the thermodynamic functions of the solution, the dissolution process is favored by the entropy of the solution, and the main contributor to the Gibbs energy is the enthalpy; when evaluating the thermodynamic mixing functions, the addition of 1-PrOH leads to an increase in the solute-solvent interactions, despite an increase in the mixing enthalpy, a factor unfavorable to the process. However, the increase in mixing entropy favors the process. This strong step-entropic compensation can be corroborated with the compensation plots, where a linear relationship between the enthalpy and the entropy of solution is observed.

By identifying the influence of MeCN as a cosolvent and that of 1-PrOH as an antisolvent, this information can be used in precipitation and crystallization processes. In addition, by having specific solubility data, the development of chromatographic methods to quantify the drug in different industrial processes can be made more efficient. Finally, the solubility data in 1-PrOH are highly relevant for the development of pharmaceutical products for topical use. Finally, this work is an important contribution to the understanding of the possible mechanisms involved in the solution process, especially in structurally related solvents and drugs.

## Figures and Tables

**Figure 2 pharmaceuticals-17-01594-f002:**
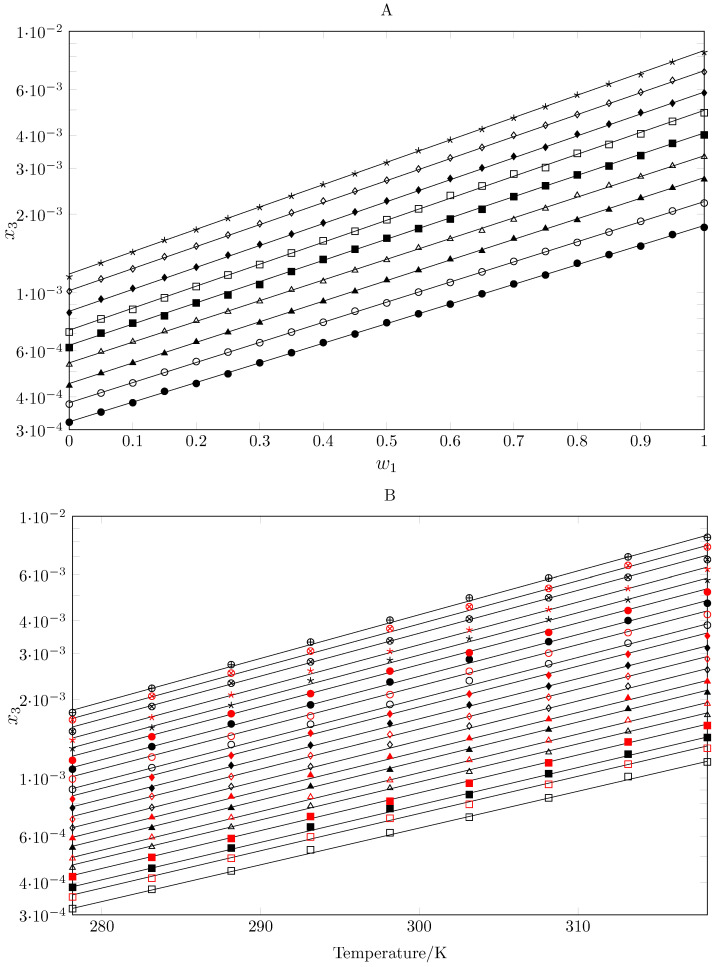
(**A**) Logarithmic mole fraction solubility of SMT ($x_{3}$) as function of the mass fraction of MeCN (1) in {MeCN (1) + 1-PrOH (2)} mixtures at different temperatures (in Kelvin). •: 278.15; ∘: 283.15; ▲: 288.15; △: 293.15; ■: 298.15; □: 303.15; ⧫: 308.15; ◊: 313.15; ★: 318.15. (**B**) Logarithmic mole fraction solubility of SMT ($x_{3}$) in different {MeCN (1) + 1-PrOH (2)} mixtures compositions ($w_{1})$ as function of temperature. □: 0.00; 
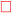
: 0.05; ■: 0.10; 
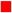
: 0.15; △: 0.20; 
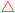
: 0.25; ▲: 0.30; 
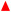
: 0.55; ◊: 0.20; 
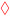
: 0.25; ⧫: 0.30; 
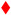
: 0.55; ∘: 0.60; 

: 0.65; •: 0.70; 

: 0.75; ★: 0.80; 

: 0.85; ⊗: 0.90; 
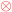
: 0.95; ⊕: 1.00.

**Figure 5 pharmaceuticals-17-01594-f005:**
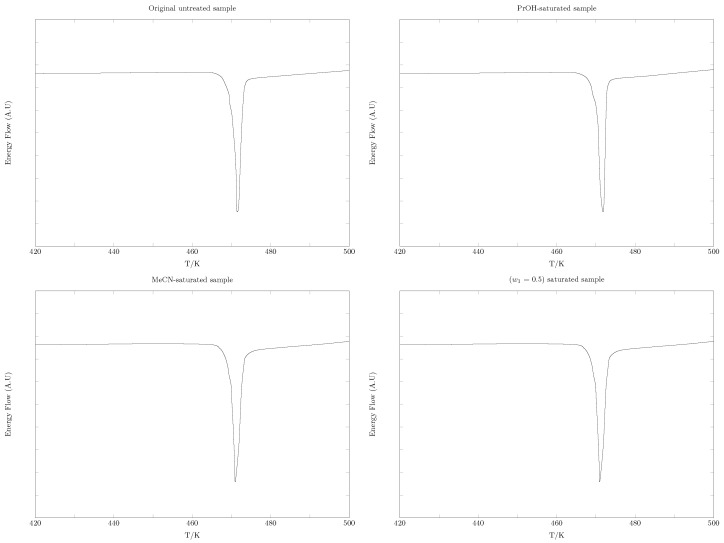
DSC thermograms of SMT as original sample and phases in equilibrium with saturated solutions. With $T_{\mathrm{fus}}$/K and $\Delta_{\mathrm{fus}}H^{\circ }$/kJ·mol^−1^ values and standard uncertainties: Original untreated sample ($T_{\mathrm{fus}}$/K = 468.5 ± 0.5, $\Delta_{\mathrm{fus}}H^{\circ }$/kJ·mol^−1^ = 33.57 ± 0.3), 1-PrOH-saturated sample ($T_{\mathrm{fus}}$/K = 468.5 ± 0.5, $\Delta_{\mathrm{fus}}H^{\circ }$/kJ·mol^−1^ = 33.8 ± 0.3), MeCN-saturated sample ($T_{\mathrm{fus}}$/K = 468.7 ± 0.5, $\Delta_{\mathrm{fus}}H^{\circ }$/kJ·mol^−1^ = 34.27 ± 0.3), and {MeCN (1) + 1-PrOH (2)} ($w_{1}$ = 0.5) saturated sample ($T_{\mathrm{fus}}$/K = 468.3 ± 0.5, $\Delta_{\mathrm{fus}}H^{\circ }$/kJ·mol^−1^ = 34.1 ± 0.3).

**Figure 6 pharmaceuticals-17-01594-f006:**
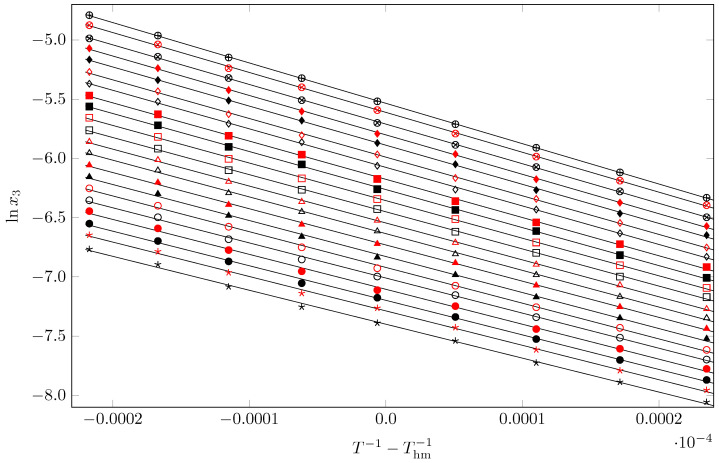
Van ’t Hoff plot of the mole fraction solubility of SMT ($x_{3}$) in different MeCN (1) + 1-PrOH (2) mixtures compositions. ★: $w_{1}$ = 0.00 (neat 1-PrOH); 

: 0.05; •: 0.10; 

: 0.15; ∘: 0.20 

: 0.25; ▲: 0.30; 
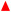
: 0.35; △: 0.40; 
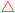
: 0.45; □: 0.50; 
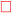
: 0.55; ■: 0.60; 
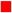
: 0.65; ◊: 0.70; 
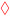
: 0.75; ⧫: 0.80; 
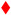
: 0.85; ⊗: 0.90; 
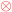
: 0.95; ⊕: 1.00 (neat MeCN).

**Figure 7 pharmaceuticals-17-01594-f007:**
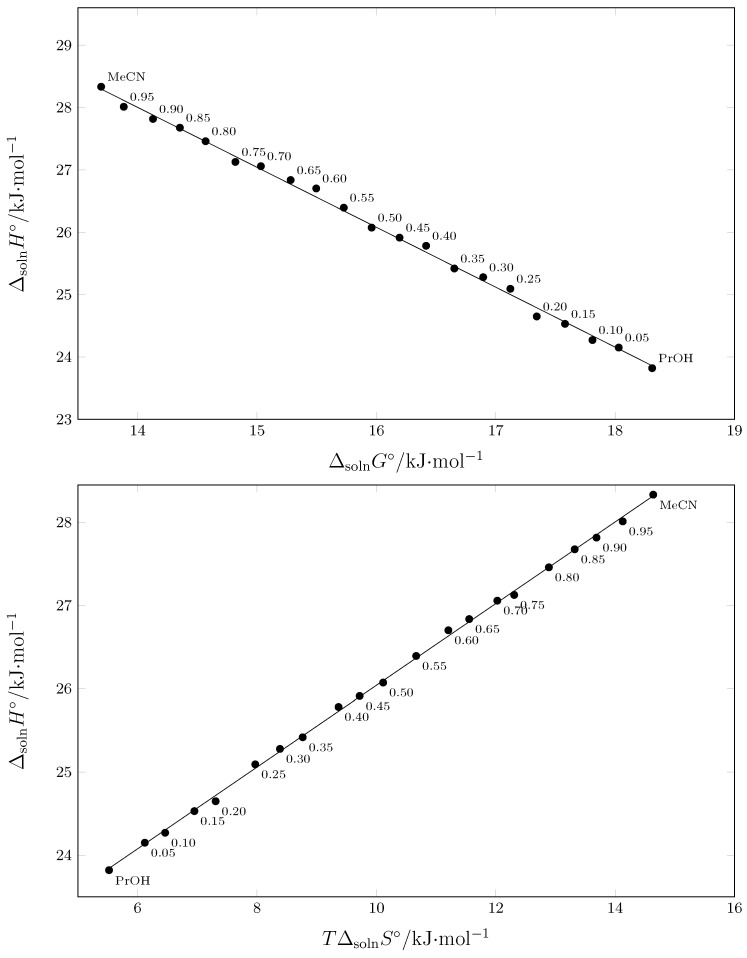
Enthalpy–entropy compensation plots for the solubility of SMT (3) in {MeCN (1) + 1-PrOH (2)} mixtures at $T_{\mathrm{hm}}$/K = 297.6 K. The points represent the mass fraction of MeCN (1) in the {MeCN (1) + 1-PrOH (2)} mixtures in the absence of SMT (3).

**Table 3 pharmaceuticals-17-01594-t003:** Apparent thermodynamic functions relative to dissolution processes of SMT (3) in {MeCN (1) + 1-PrOH (2)} mixtures at $T_{\mathrm{hm}}$/K = 297.6 as a function the mass fraction of MeCN (1) ($w_{1}$) in the {MeCN (1) + EtOH (2)} mixtures free of SMR (3).

*w*_1_ ^a^	$\Delta_{\mathrm{soln}}G^{\circ }$ (kJ·mol^−1^) ^b^	$\Delta_{\mathrm{soln}}H^{\circ }$ (kJ·mol^−1^) ^b^	$\Delta_{\mathrm{soln}}S^{\circ }$ (J·mol^−1^·K^−1^) ^b^	$T_{\mathrm{hm}}\Delta_{\mathrm{soln}}S^{\circ }$ (kJ·mol^−1^) ^b^	$\zeta_{H}$	$\zeta_{\mathrm{TS}}$
0.00	18.31	23.82	18.54	5.52	0.81	0.19
0.05	18.03	24.15	20.58	6.12	0.80	0.20
0.10	17.81	24.27	21.70	6.46	0.79	0.21
0.15	17.58	24.53	23.34	6.95	0.78	0.22
0.20	17.34	24.65	24.55	7.31	0.77	0.23
0.25	17.12	25.09	26.78	7.97	0.76	0.24
0.30	16.89	25.28	28.17	8.38	0.75	0.25
0.35	16.65	25.42	29.45	8.77	0.74	0.26
0.40	16.42	25.78	31.47	9.37	0.73	0.27
0.45	16.19	25.91	32.66	9.72	0.73	0.27
0.50	15.96	26.07	33.98	10.11	0.72	0.28
0.55	15.73	26.39	35.84	10.67	0.71	0.29
0.60	15.50	26.70	37.65	11.21	0.70	0.30
0.65	15.28	26.84	38.83	11.55	0.70	0.30
0.70	15.03	27.06	40.41	12.03	0.69	0.31
0.75	14.82	27.13	41.36	12.31	0.69	0.31
0.80	14.57	27.46	43.31	12.89	0.68	0.32
0.85	14.35	27.68	44.77	13.32	0.68	0.33
0.90	14.13	27.82	46.00	13.69	0.67	0.33
0.95	13.88	28.01	47.47	14.13	0.67	0.34
1.00	13.69	28.33	49.19	14.64	0.66	0.34
Ideal	9.89	21.50	39.02	11.61	0.649	0.351

^a^ $u_{r}$($w_{1}$) = 0.0010, *u*(*T*) = 0.10 K. ^b^ $u_{r}$($\Delta_{\mathrm{soln}}G^{\circ }$) = 0.030, $u_{r}$($\Delta_{\mathrm{soln}}H^{\circ }$) = 0.050, $u_{r}$($\Delta_{\mathrm{soln}}S^{\circ }$) = 0.058, $u_{r}$($T\Delta_{\mathrm{soln}}S^{\circ }$) = 0.058.

**Table 4 pharmaceuticals-17-01594-t004:** Apparent thermodynamic functions relative to mixing processes of SMT (3) in {MeCN (1) + 1-PrOH (2)} mixtures at $T_{\mathrm{hm}}$/K = 297.6 as a function the mass fraction of MeCN (1) ($w_{1}$) in the {MeCN (1) + EtOH (2)} mixtures free of SMR (3).

$w_{1}$ ^a^	$\Delta_{\mathrm{mix}}G^{\circ }$ (kJ·mol^−1^) ^b^	$\Delta_{\mathrm{mix}}H^{\circ }$ (kJ·mol^−1^) ^b^	$\Delta_{\mathrm{mix}}S^{\circ }$ (J·mol^−1^·K^−1^) ^b^	$T\Delta_{\mathrm{mix}}S^{\circ }$ (kJ·mol^−1^) ^b^
0.00	8.42	2.33	−20.48	−6.09
0.05	8.14	2.66	−18.44	−5.49
0.10	7.92	2.77	−17.32	−5.15
0.15	7.69	3.03	−15.68	−4.67
0.20	7.46	3.15	−14.47	−4.31
0.25	7.24	3.60	−12.24	−3.64
0.30	7.01	3.78	−10.85	−3.23
0.35	6.77	3.92	−9.57	−2.85
0.40	6.53	4.28	−7.55	−2.25
0.45	6.31	4.42	−6.36	−1.89
0.50	6.07	4.58	−5.04	−1.5
0.55	5.84	4.90	−3.18	−0.95
0.60	5.61	5.20	−1.36	−0.41
0.65	5.40	5.34	−0.19	−0.06
0.70	5.15	5.56	1.39	0.41
0.75	4.93	5.63	2.34	0.70
0.80	4.68	5.96	4.29	1.28
0.85	4.47	6.18	5.75	1.71
0.90	4.24	6.32	6.98	2.08
0.95	4.00	6.51	8.45	2.52
1.00^**c**^	3.81	6.83	10.17	3.03

^a^ $u_{r}$($w_{1}$) = 0.0010; *u*(*T*) = 0.10 K. ^b^ $u_{r}$($\Delta_{\mathrm{mix}}G^{\circ }$) = 0.037, $u_{r}$($\Delta_{\mathrm{mix}}H^{\circ }$) = 0.050, $u_{r}$($\Delta_{\mathrm{mix}}S^{\circ }$) = 0.063, $u_{r}$($T\Delta_{\mathrm{mix}}S^{\circ }$) = 0.063.

**Table 5 pharmaceuticals-17-01594-t005:** Chemical reagents used in research development.

Chemical Name	CAS ^a^	Purity in Mass Fraction	Analytic Technique ^b^
SMT ^c^	127-79-7	>0.990	HPLC
EtOH ^c^	64-17-5	0.998	GC
1-PrOH ^d^	71-23-8	0.998	GC
MeCN ^d^	75-05-8	0.998	GC

^a^ Chemical Abstracts Service Registry Number. ^b^ HPLC is high-performance liquid chromatography; GC is gas chromatography. ^c^ Sigma-Aldrich, Burlington, MA, USA. ^d^ Supelco, Burlington, MA, USA.

## Data Availability

Data are contained within the article.

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
