# Peer review of "Dissolution Thermodynamics of the Solubility of Sulfamethazine in (Acetonitrile + 1-Propanol) Mixtures"

_pharmaceuticals, 2024, doi:10.3390/ph17121594_

Round 1

Reviewer 1 Report

Comments and Suggestions for Authors

This manuscript describes a very thorough analysis of the solubilities of sulfamethazine in acetonitrile/1-propanol mixtures. The study is a continuation of such studies of sulphadiazine and sulphamerazine in solvent mixtures.  The results are potentially useful for the pharmaceutical industry in the synthesis and\ purification of such compounds.  What is missing is an analysis of these compounds in the solution phase.  There have been some reports that compounds do not dissolve completely (see Brian Shoichet) in solvents thereby preventing accurate dosages.  It would be useful to add such an analysis to this manuscript for completeness.

The other analyses of compound solubility in solvent mixtures have been published in journals such as Molecules, Fluid Phase Equilibria and J. Chem. Thermodynamics.  
This study also belongs into such a journal rather than Pharmaceuticals. 

Author Response

We appreciate the reviewer's comments, but all the data reported are on saturation concentrations (solution) and precisely solve problems such as those mentioned (there have been some reports that compounds do not completely dissolve (see Brian Shoichet) in solvents, thus preventing accurate dosing).

On the other hand, the special issue in which the manuscript is presented invites the submission of manuscripts on the topics of solubilization, solubility, and poorly water-soluble drugs. On the other hand, our group has published an article in this research area in this journal; therefore, we believe that the current manuscript not only meets the parameters of the journal, but also falls within its publication topics

Reviewer 2 Report

Comments and Suggestions for Authors

The study of Delgado et al. is a comprehensive investigation of the solubility behavior of sulfamethazine in mixtures of ACN and 1-propanol. The authors measured the solubility at 9 temperatures and disclose the apparent dissolution and mixing thermodynamics, addressed the activity coefficient issues.

The results of the study were obtained for the first time.

The authors proved the importance of the solvents used for pharmaceuticals. The Introduction provides sufficient information on the reported literature. At the same time, the reference list contains too much old literature that, to my opinion could be replaced with the more actual to date. The paper is well-illustrated and structured.

The manuscript meets the scope of the Pharmaceuticals but the authors should address the comments below to improve the quality of the manuscript before publication:

Abstract

1) Lines 4-6

The sentence needs reconsideration.

2) Lines 8-10

The sentence needs reconsideration.

3) In order to understand the significance of the "most relevant results" (Lines 8-10), the range of the examined temperatures needs to be indicated in Line 5.

4) Additionally, the solubility regularities in mixed solvents are also should be indicated in the abstract.

5) Line 13

"…a possible relationship…" Why "possible"?

Introduction

6) Lines 25-29

For better comprehension it would be better to split the sentence.

Results

7) The equation for x3 calculation is required. Provide, please.

8) Line 64

"Results" should be "Results and Discussion"

9) Line 67

ω1 should be specified in the text, not only in the footnote to Tables.

10) Lines 69-72

Check the sentence, please.

11) Lines 86-87

Abbreviations of sulfadiazine and sulfamerazine should be provided in parentheses just after the names of drugs.

12) Line 97

The references for cosolvent mixtures MeCN (1) + MeOH (2)and {MeCN (1) + 1-PrOH (2)} are required. The same in the caption to Fig. 4.

13) After Line 105

Explanation of the solubility in MeOH is desirable.

Materials and Methods

3.1. Reagents

14) It is necessary to specify whether the reagents were used in their original state or were subjected to additional purification.

15) Line 223

"…the shake flask method. Higuchi and Connors [52–54]."

Please, replace with "…the shake flask method according to Higuchi and Connors [52–54]."

16) Lines 229-231

It is unclear, at what point thermodynamic equilibrium is reached in the solid phase-solution system and how this is confirmed. Specify, please.

17) Lines 236-238

Filtration. The issue of supersaturation is crucial for correct determination of the solubility in the case of shake-flask procedure. How was it confirmed that the examined solution was not supersaturated?

18) Line 238

"… 0.45 μm membrane. μm membranes…" - correct, please

19) Line 240-242

An important question: What about the calibration curves at different ratio of the tested solvents?

What about λmax in ACN and 1-propanol?

20) The reference list contains too much old literature. Could the authors replace some of them with newer ones?

Author Response

1) Lines 4-6. The sentence needs reconsideration.

Reply: The word “widely” was removed. It may sound like hyperbole.

2) Lines 8-10. The sentence needs reconsideration.

Reply: Totally agree with the evaluator, no relevant data was really presented. Data was presented showing a different behavior to that reported for aqueous solutions.

3) In order to understand the significance of the "most relevant results" (Lines 8-10), the range of the examined temperatures needs to be indicated in Line 5.

Reply: The 9 study temperatures are now presented

4) Additionally, the solubility regularities in mixed solvents are also should be indicated in the abstract.

Reply: The regularity of SMT solubility has been described.

5) Line 13

"…a possible relationship…" Why "possible"?

Reply: The paragraph has been removed as it is better covered in the previous paragraph.

6) Lines 25-29

For better comprehension it would be better to split the sentence.

Reply: The paragraph has been rewritten for clarity.

7) The equation for x3 calculation is required. Provide, please.

Reply: The following equation is presented

8) Line 64

Reply: The following equation is presented

Reply: Corrected

9) Line 67

ωshould be specified in the text, not only in the footnote to Tables.

Reply: It has been indicated in line 78

10) Lines 69-72

Check the sentence, please.

Reply: Thank you for your careful reading of the manuscript, the error has been corrected. 

11) Lines 86-87

Abbreviations of sulfadiazine and sulfamerazine should be provided in parentheses just after the names of drugs.

Reply: Lines 18, 98 and 99

12) Line 97

The references for cosolvent mixtures MeCN (1) + MeOH (2)and {MeCN (1) + 1-PrOH (2)} are required. The same in the caption to Fig. 4.

Reply: Reference 19 has been added in the text, the data presented in Figure 4 correspond to those reported in reference 19 and the experimental data reported in this paper.

13) After Line 105

Explanation of the solubility in MeOH is desirable.

Reply: It is added at the end of Section 2.1.

14) It is necessary to specify whether the reagents were used in their original state or were subjected to additional purification.

Reply: Reagents were used in their original condition without further purification.

15) Line 223

"…the shake flask method. Higuchi and Connors [52–54]."

Please, replace with "…the shake flask method according to Higuchi and Connors [52–54]."

Reply: We welcome the proposed amendment

16) Lines 229-231

It is unclear, at what point thermodynamic equilibrium is reached in the solid phase-solution system and how this is confirmed. Specify, please.

Reply: The procedure for ensuring thermodynamic equilibrium is explained and reported.

17) Lines 236-238

Filtration. The issue of supersaturation is crucial for correct determination of the solubility in the case of shake-flask procedure. How was it confirmed that the examined solution was not supersaturated?

Reply: Please see section Thermodynamic equilibrium

18) Line 238

"… 0.45 μm membrane. μm membranes…" - correct, please

19) Line 240-242

Reply: corrected

19) Line 240-242

An important question: What about the calibration curves at different ratio of the tested solvents?

What about λmax in ACN and 1-propanol?

Reply: Both calibration curves and dilutions are performed in absolute ethanol due to the high solubility of SMT in this solvent. As the sample dilution is in most cases greater than 1:100, the effect of the solvents in the mixtures (MeCN and 1-PrOH) on the linearity or wavelength of maximum absorbance is negligible

20) The reference list contains too much old literature. Could the authors replace some of them with newer ones?

Reply: Some current references were added

Reviewer 3 Report

Comments and Suggestions for Authors

The manuscript entitled “Dissolution thermodynamics of the solubility of sulfamethazine in (acetonitrile +1-propanol) mixtures” shows interesting basic-knowledge for solubilizing sulfamethazine by using mixtures of acetonitrile and propanol.  The authors determined the optimum mass fractions of such solvents for facilitating the solubility of sulfamethazine at various temperatures and quantified the concentrations of the drug that was dissolved in the mixture solvents by using UV/Vis spectrophotometer.  However, there are some comments and questions for this manuscript as follows:

1. The authors have to use a word “sulfamethazine/sulphamethazine” consistently throughout the manuscript.  Please carefully check and correct.

2. Before using abbreviations, the authors have to present their full word first and provide their abbreviation in the brackets after their full word, for example, SMT, SMR, SD. Then, the authors should use these abbreviations throughout the manuscript when they were mentioned again.

3. The authors should indicate the novelties and research gaps of this work in the Introduction section of this manuscript.

4. This manuscript will be more interesting if the authors propose the possibility benefits of the obtained knowledge to the real situations in the Pharmaceutical Industries.

5. In Table 1, and 2, the authors have to inform what the data shown in these tables were.  Please add the name of the presented parameter in the head of each table. For example, the mole fraction solubility of sulfamethazine (x104) at various temperatures (K): unit…….

6. Please differentiate the DSC thermograms in Fig. 5 and inform what they were.

7.  The authors should specify the objectives of the DSC study of SMT solid phase.  Since the solid sulfamethazine for this study was from the bottom phase of the drug that was not dissolved in the solvent.  Therefore, it was not surprised to obtain the consistent DSC thermograms of the pure sulfamethazine and the excess sulfamethazine located on the bottom of flasks as they still were drug powder of the original.

8. What is the meaning of “other solvent systems” in the last sentences of Page 11 (line200)? Could the authors specify what they were? Since the solvents used in this study were acetonitrile, propanol, and their mixtures.

9. The authors should indicate the applications of the obtained results from this study for drug product development, production, and quality control in the Pharmaceutical Industries and add into the Conclusion section of this manuscript.

Author Response

  1. The authors have to use a word “sulfamethazine/sulphamethazine” consistently throughout the manuscript.  Please carefully check and correct.

Reply: Thanks for the comment, the word Sulphamethazine has been changed to Sulfamethazine throughout the document.

  1. Before using abbreviations, the authors have to present their full word first and provide their abbreviation in the brackets after their full word, for example, SMT, SMR, SD. Then, the authors should use these abbreviations throughout the manuscript when they were mentioned again.

Reply: Thanks for your suggestion, changes have been made throughout the document.

  1. The authors should indicate the novelties and research gaps of this work in the Introduction section of this manuscript.

Reply: The knowledge gap and the impact of the research were better described.

  1. This manuscript will be more interesting if the authors propose the possibility benefits of the obtained knowledge to the real situations in the Pharmaceutical Industries.

Reply: Please see paragraph 31-36

  1. In Table 1, and 2, the authors have to inform what the data shown in these tables were.  Please add the name of the presented parameter in the head of each table. For example, the mole fraction solubility of sulfamethazine (x104) at various temperatures (K): unit…….

Reply: The temperature units were added

  1. Please differentiate the DSC thermograms in Fig. 5 and inform what they were.

Reply: Figure 5 has been changed

  1. The authors should specify the objectives of the DSC study of SMT solid phase.  Since the solid sulfamethazine for this study was from the bottom phase of the drug that was not dissolved in the solvent.  Therefore, it was not surprised to obtain the consistent DSC thermograms of the pure sulfamethazine and the excess sulfamethazine located on the bottom of flasks as they still were drug powder of the original.

Reply: This is a very precise observation, so in section 3.2 points 2 and 5 the procedure, the aim of the DSC and the procedure for obtaining crystals from the saturated solution are described. However, it is clear that what the reviewer mentioned may occur, so we try to evaluate the solid phase in equilibrium by saturating the mixtures at a higher temperature and then lowering the temperature to obtain some crystals by precipitation.

  1. What is the meaning of “other solvent systems” in the last sentences of Page 11 (line200)? Could the authors specify what they were? Since the solvents used in this study were acetonitrile, propanol, and their mixtures.

Reply: The idea has been rewritten to be clearer about the mixtures in question.

  1. The authors should indicate the applications of the obtained results from this study for drug product development, production, and quality control in the Pharmaceutical Industries and add into the Conclusion section of this manuscript.

Reply: A paragraph has been added to the conclusions emphasizing the importance of research in this area.

Round 2

Reviewer 1 Report

Comments and Suggestions for Authors

The issue of solubility has been addressed significantly by Shoichet where his group found that many compounds form clusters in solution that are not available to the targets.  Although the solution looks clear, it is not.  This should be followed up with these compounds since their efficacy depends on availability in solution.

Reviewer 2 Report

Comments and Suggestions for Authors

The authors made the corrections. The manuscript can be published.

Reviewer 3 Report

Comments and Suggestions for Authors

The authors responded to all addressed comments and questions with acceptable reasons.  Therefore, this manuscript could be accepted for publishing in the journal in the present form.